# Patient-Related Risk Factors for Unplanned 30-Day Hospital Readmission Following Primary and Revision Total Knee Arthroplasty: A Systematic Review and Meta-Analysis

**DOI:** 10.3390/jcm10010134

**Published:** 2021-01-02

**Authors:** Daniel Gould, Michelle M Dowsey, Tim Spelman, Olivia Jo, Wassif Kabir, Jason Trieu, James Bailey, Samantha Bunzli, Peter Choong

**Affiliations:** 1Department of Surgery, University of Melbourne, St. Vincent’s Hospital Melbourne, 3065 Melbourne, Australia; mmdowsey@unimelb.edu.au (M.M.D.); tim@burnet.edu.au (T.S.); jo.olivia1310@gmail.com (O.J.); kabir.wassif@gmail.com (W.K.); jtrieumd@gmail.com (J.T.); samantha.bunzli@unimelb.edu.au (S.B.); pchoong@unimelb.edu.au (P.C.); 2Department of Othopaedics, St. Vincent’s Hospital Melbourne, 3065 Melbourne, Australia; 3School of Computing and Information Systems, University of Melbourne, 3052 Melbourne, Australia; baileyj@unimelb.edu.au

**Keywords:** readmission, arthroplasty, knee, risk, patient, prognosis

## Abstract

Total knee arthroplasty (TKA) is a highly effective procedure for advanced osteoarthritis of the knee. Thirty-day hospital readmission is an adverse outcome related to complications, which can be mitigated by identifying associated risk factors. We aimed to identify patient-related characteristics associated with unplanned 30-day readmission following TKA, and to determine the effect size of the association between these risk factors and unplanned 30-day readmission. We searched MEDLINE and EMBASE from inception to 8 September 2020 for English language articles. Reference lists of included articles were searched for additional literature. Patients of interest were TKA recipients (primary and revision) compared for 30-day readmission to any institution, due to any cause, based on patient risk factors; case series were excluded. Two reviewers independently extracted data and carried out critical appraisal. In-hospital complications during the index admission were the strongest risk factors for 30-day readmission in both primary and revision TKA patients, suggesting discharge planning to include closer post-discharge monitoring to prevent avoidable readmission may be warranted. Further research could determine whether closer monitoring post-discharge would prevent unplanned but avoidable readmissions. Increased comorbidity burden correlated with increased risk, as did specific comorbidities. Body mass index was not strongly correlated with readmission risk. Demographic risk factors included low socioeconomic status, but the impact of age on readmission risk was less clear. These risk factors can also be included in predictive models for 30-day readmission in TKA patients to identify high-risk patients as part of risk reduction programs.

## 1. Introduction

Total knee arthroplasty (TKA) is a highly effective treatment for advanced osteoarthritis of the knee [1,2]. The number of procedures being performed each year continues to grow [3]. However, despite its widespread success and increasing utilisation rates, a range of complications can arise following TKA surgery [4], some of which necessitate hospital readmission. Hospital readmission following TKA is often a marker of surgical complications [5,6]. Post-operative hospital readmissions are recognised as a significant cause for concern because such events disrupt the patient’s postoperative recovery, incur significant costs to the healthcare system¸ and controversially have been used as a marker of quality of care [7,8,9]. In the USA, unplanned hospital readmissions for all patient populations overall are estimated to cost over USD 17 billion [10]. In 2004, Germany became the first country to implement financial incentives to reduce readmissions [11]. Five years later, in 2009, the United States’ Centers for Medicare & Medicaid Services introduced the Hospital Readmissions Reduction Program [9] and expanded this program in 2014 to include TKA [12,13], where hospital penalties are applied for readmission signalling government authorities to view readmission as an issue requiring attention.

By improving our ability to accurately predict individual patients’ risk of readmission following TKA, we may reduce the rate of avoidable readmissions through optimising shared decision-making and consent processes; better preparing patients for surgery and developing personalised management and discharge planning for higher risk patients. Furthermore, some readmissions are potentially avoidable and can be targeted through the development of personalised management and discharge planning for higher risk patients. Patients whose risk of readmission is difficult to mitigate, such as those patients with non-modifiable risk factors, can still benefit from risk prediction because they will be more informed as they work with the surgeon to weigh the potential benefits and risks of the procedure, and patients who have more realistic expectations regarding their likely postoperative course are more likely to achieve a satisfactory outcome following TKA [14].

Our previous narrative literature review has explored patient-related risk factors for 30-day readmissions in TKA [15]. Comorbidities such as diabetes, chronic kidney disease, and bleeding disorders were consistently shown to increase risk, but demographic factors such as age, sex, and body mass index (BMI) demonstrated a more complex and less consistent influence on 30-day readmission. Using age as an example, some researchers dichotomise at a particular number, while others create multiple categories at different cut-points with different interval durations, such as five years or 10 years. The inconsistent way different researchers categorise predictor variables when modelling risk, and the increasing age, BMI, and physical activity demands of the TKA patient population over time [16,17,18], may contribute and confound this further. Thus, the purpose of this systematic review was to apply rigorous systematic review methodology to synthesise the evidence pertaining to patient risk factors for 30-day readmission following TKA. Contrasting the more subjective style of the narrative review, which identified broad themes in the literature, the current work synthesises the evidence in the most objective manner possible, including: critical appraisal of all included articles, quantitative synthesis when appropriate, narrative synthesis when quantitative synthesis was not possible, and summary of findings in accordance with a version of the Grading of Recommendations Assessment, Development and Evaluation (GRADE) approach modified for systematic reviews of prognostic factors [19].

### Objectives

The objectives of this review were to (1) identify patient-related characteristics that are associated with increased risk of unplanned 30-day readmission following TKA and (2) determine the effect size of the association between the identified risk factors and unplanned 30-day readmission [20]. This systematic review and meta-analysis synthesises existing knowledge, determines the effect size of identified factors, and aims to resolve uncertainty when discrepancies arise between reports.

## 2. Methods

### 2.1. Protocol and Registration

The protocol for this systematic review has been published [20] and registered with the International Prospective Register of Systematic Reviews (PROSPERO-CRD42019118154). Protocol deviations, and justifications for them, are outlined in Appendix A. Deviations were minor and did not alter the objectives, nor the direction of this systematic review. This review was conducted in accordance with the Preferred Reporting Items for Systematic Reviews and Meta-Analyses (PRISMA) statement [21].

### 2.2. Eligibility Criteria

We used the adapted Patient, Intervention, Comparator, Outcome (PICO) framework [22] to answer the following questions: which patient-related factors confer increased risk of unplanned 30-day readmission following TKA, and to what extent do these factors influence the risk? Eligible studies involved TKA recipients (primary and revision) compared for 30-day readmission to any institution, due to any cause, on the basis of patient risk factors (presence vs absence of each risk factor). Case series were excluded, but all other types of quantitative study design were eligible for inclusion, including retrospective and observational studies.

No restrictions were placed on the date of publication for this review.

### 2.3. Information Sources, Search Strategy, and Study Inclusion

MEDLINE and EMBASE were searched from inception to 5 February 2020, using search terms related to total knee arthroplasty and patient-related risk factors for 30-day readmission. An experienced research librarian was consulted to assist with development of the search strategy, and no restriction was placed on grey literature. The full search strategy for both databases is available in Appendix A. The search was repeated immediately prior to final analysis (8th September 2020) to obtain studies that had been published since initially searching the database. Reference lists of articles included after full text screening were also screened.

The search strategy was carried out in line with the protocol. In brief, two reviewers (DG and OJ) pilot-tested the eligibility criteria on a 10% sample of titles and abstracts of articles obtained from the database search. A third reviewer (MD) arbitrated discussions between DG and OJ to resolve any major difference in opinion arising from this process. The eligibility criteria were then applied to titles and abstracts and, subsequently, to the full text of articles that were deemed eligible for inclusion following initial screening.

Authors of articles potentially eligible for inclusion were contacted to provide the requisite data for inclusion.

### 2.4. Data Collection Process

Two reviewers (DG and JT) independently extracted data using a standardised data extraction form. The process was pilot-tested on 10 articles from the final set of included articles following full-text screening. Any points of clarification were resolved by discussion between DG and JT. DG identified additional information required in order to complete the critical appraisal assessment and this was verified by JT as well as the reviewer who completed the critical appraisal assessment with DG (WK).

### 2.5. Data Items

Data items extracted were those listed in the protocol, as well as several additional items necessary for critical appraisal (Appendix A).

As outlined in the protocol amendments (Appendix A), adjusted odds ratios (ORs) were preferentially collected over risk ratios (RRs), as this reflects how data were reported in the vast majority of included studies, which were retrospective in nature. As the estimated 30-day readmission rate for TKA ranges from 3% [5] to 4.6% [23] and this is well below the 10% threshold whereby the OR provides a reasonable approximation of the RR [20,24], it is unlikely that this would have altered the key findings.

### 2.6. Risk of Bias of Individual Studies

The Joanna Briggs Institute (JBI) critical appraisal tool [25] was used for all studies. A semi-quantitative method was used, based on that used by Goplen et al. [26] in a systematic review on outcomes in total joint arthroplasty (TJA) patients. However, since there is no valid cut-off value to determine “high” or “low” methodological quality, we assessed quality in relative terms by splitting the studies according to risk of bias quartiles. The item: “Were the groups/participants free of the outcome at the start of the study (or at the moment of exposure)?” was omitted from the checklist as it is not possible for a patient to be readmitted prior to their TKA procedure. The remaining 10 items were given equal weighting with a higher score indicating greater risk of bias. Each “no” was given a score of 2, each ‘unclear a score of 1, and each “yes” a score of 0. Arbitrarily selecting a cut-point above which a study is considered to be at high risk of bias is problematic because there are no clear guidelines pertaining to the choice of such a cut-point. Instead, quartiles of risk of bias were calculated to distinguish between higher quality studies (lower risk of bias quartiles) and lower quality studies (higher risk of bias quartiles). This use of quartiles is similar to that used by Detweiler et al., 2016 [27] in their exploration of methodological quality of systematic reviews, and we believe it provides readers with an easily interpretable way of judging the relative methodological quality of included studies without generating an overwhelming number of categories.

Selective reporting was also assessed for each study. This was included in the summary table for critical appraisal but because it was not a JBI checklist item, it did not contribute to the calculation of risk of bias score.

Table 1 depicts a generic example of the way in which the outcome of critical appraisal is presented in this review for an imaginary study, for the sake of illustration.

### 2.7. Risk of Bias across Studies

Risk of bias across studies, i.e., publication bias, was minimised by using a comprehensive systematic literature search strategy and placing no restriction on grey literature.

Although articles in languages other than English were excluded, in order to determine if this created a potential source of bias, those with available English language abstracts were screened to determine the likelihood of their inclusion if the full text was available in English.

### 2.8. Synthesis of Results

Summary of findings tables were constructed, according to the modified Grading of Recommendations Assessment, Development and Evaluation (GRADE) approach [19], for each of two categories of patient-related risk factors: comorbidities and demographics. Variables that did not fit into either of these categories were summarised in a separate summary of findings table. A summary of findings table was also produced for the studies that reported on revision TKA patients (see Appendix A). For each risk factor, we considered the seven GRADE criteria [19] (study limitations, inconsistency, indirectness, imprecision, publication bias, moderate/large effect size, dose effect) in addition to phase of study (phase 1 = explanatory research aimed to identify associations between potential prognostic factors and the outcome; phase 2 = explanatory research aimed to confirm independent associations between potential prognostic factors and the outcome; phase 3 = research aimed to understand prognostic pathways). For each of these eight criteria, a score of one was given when there was no serious limitation identified, and a score of zero when a serious limitation was identified. A score of <3 was considered very low quality = very little confidence in the effect estimate: true effect likely to be substantially different from the estimate of effect; a score of 3 was considered low quality = confidence in the effect estimate is limited: the true effect may be substantially different from the estimate of the study; a score of 4 or 5 was considered moderate quality = moderately confident in the effect estimate: true effect is likely to be close to the estimate of the effect, but there is a possibility that it is substantially different; a score of >5 was considered high quality = very confident that the true effect lies close to that of the estimate of the effect.

A database can be analysed in different studies, such that multiple papers included in this review report findings from analyses on the same cohort of patients, giving rise to the potential for sample dependence. In such cases, when these studies had overlapping time periods (e.g., one study analysed the 2011–2014 cohort and another the 2011–2012 cohort), the study with the longest period of data collection was selected as the representative study for that cohort, for each variable. When two or more studies had the same data collection period, the study with higher methodological quality was selected as the representative study. Although only these representative studies were included in the summary of findings tables, data from all studies are available in Appendix A.

Table 2 depicts a generic example of the way in which the summary of findings is presented in this review for some imaginary studies and risk factors, for the sake of illustration. For the narrative synthesis, + indicates the risk factor increased risk of readmission, 0 indicates it was not correlated, and – indicates it was protective against readmission.

Results from studies with unadjusted effect estimates and univariate comparisons were also included, alongside additional analyses conducted in individual studies (Appendix A).

When multiple reports on a particular risk factor from the same study cohort were identified, the article with the longest follow-up period was used as the representative study for all overlapping studies with comparable eligibility criteria, but reference was made to all overlapping studies.

R statistical software [28] (version 3.5.3 (2019-03-11)—“Great Truth”) was used for meta-analysis, including the meta [29] (version 4.11-0) and tidyverse [30] (version 1.3.0) packages. The code used for meta-analysis is available in Appendix A.

### 2.9. Sensitivity Analyses Were Conducted, Based on the Following Criteria

Removal of studies with lower methodological quality, i.e., higher risk of bias.Removal of studies of mixed cohorts, i.e., a combined cohort of primary and revision TKA patients.Removal of studies with substantially different patient eligibility criteria to other studies, for example, those that restricted their analysis to patients over 80 years old where other studies placed no restriction on age.

If the I^2^ score exceeded 60% on sensitivity analysis, the variable was analysed qualitatively through narrative synthesis.

## 3. Results

### 3.1. Study Selection

Sixty-nine studies were included in this review; 374 records underwent title and abstract screening after removal of duplicates, of which 193 underwent full-text screening to include the final 69 articles, 16 of which were included in the meta-analysis. See Figure 1 for the total number of studies screened. Table 3 presents general information pertaining to all of the included studies. Appendix A contains detailed information on a range of variables extracted from included studies. Sixty-three articles (93%) reported retrospective cohort studies, 57/69 (83%) were conducted in the USA. Thirty-three of 69 studies (48%) used the data from the American College of Surgeons’ National Surgical Quality Improvement Program (NSQIP), 13/69 (19%) used data from a single institution, and the remainder used a variety of registries and multi-hospital databases.

Three authors of potentially eligible articles were contacted to provide data necessary for inclusion, and one returned the required data [31].

### 3.2. Risk of Bias of Individual Studies

Appendix A shows that studies at 0–15% risk of bias (RoB) comprised the first RoB quartile, studies at 20% RoB comprised the second quartile, studies at 25–35% RoB comprised the third quartile, and studies at 40–70% RoB comprised the fourth quartile. Only 2/69 studies (3%) definitively addressed loss to follow-up, i.e., loss to follow-up occurs when patients are not routinely contacted during the 30-day post-discharge period to ascertain whether they were readmitted even if the readmission was to an institution other than that at which they underwent TKA, whereas three provided some level of justification and the remaining 64 studies clearly did not adequately address this. Similarly, only 14/69 studies (20%) clearly documented a strategy to deal with incomplete follow-up. Only 9/69 studies (13%) reported adjusted analyses with a clearly documented and robust method of handling missing data, whereas 49/69 studies (71%) reported adjusted analyses without mention of how missing data were handled, and 11/69 studies (16%) reported only unadjusted analyses. These 11 studies were therefore penalised for not adequately addressing confounding results. Thirty-one studies also had evidence of selective reporting.

### 3.3. Risk of Bias across Studies

Funnel plots were deemed inappropriate due to an insufficient number (*n* < 10) of compatible studies available for meta-analysis of any risk factor [98].

The only non-English-language article eligible for title and abstract screening was in German and the abstract was available in English. Based on title and abstract screening, the article did not meet the inclusion criteria for full-text screening. Therefore, exclusion of non-English studies did not affect the findings of this review.

There was strong geographical bias with most studies being conducted in the USA, as seen in Table 3. Fifty-seven studies used data from the USA, three from Canada, two from each of Denmark, Taiwan, and Singapore, and one each from Australia, Colombia, and the UK.

In accordance with Huguet et al. 2013 [19], publication bias was assumed to be a serious limitation for every prognostic factor except those that were analysed in multiple studies of different sample sizes from relatively small (<10,000 participants) to large (>10,000 participants). While publication bias could not be ruled out from these risk factors, we considered it unlikely to be a serious limitation when considering the overall weight of evidence for these factors. See the summary of findings tables (Table 4, Table 5 and Table 6, and Appendix A) for further detail on each risk factor.

### 3.4. Synthesis of Evidence

Table 4, Table 5 and Table 6, and Appendix A, present the findings for every risk factor identified in the included studies, including high, moderate, low, and very low-quality evidence. A summary of the high and moderate quality evidence for risk factors that correlated with readmission are presented. Forest plots for all meta-analyses are available (Appendix A).

#### 3.4.1. Comorbidities

On meta-analysis, risk factors associated with 30-day readmission were: arrhythmias (including atrial fibrillation), anaemia, deficiency anaemias, peripheral vascular disease, liver disease, and coagulopathy. On narrative synthesis, risk factors associated with 30-day readmission were: hypertension, congestive heart failure (CHF), diabetes, elevated preoperative international normalised ratio (INR), elevated serum blood urea nitrogen (BUN), reduced serum albumin, depression, drug abuse, history of cancer, and chronic kidney disease (CKD). On narrative synthesis, increasing comorbidity burden as indicated by Charlson Comorbidity Index, Elishauser Index, and Diagnosis-Related Group roughly correlated with increased readmission risk. This relationship was not observed strongly for American College of Anaesthesiologists (ASA) Classification.

Body mass index (BMI) was categorised in different ways in various studies, but on meta-analysis none of these categories were correlated with readmission. Similarly, when BMI was presented as a continuous variable it did not consistently correlate with readmission. However, when presented in categories, both obesity and morbid obesity did correlate with readmission both on meta-analysis and on narrative synthesis. Note that obesity categories were not clearly defined in these studies which used the labels ‘obesity’ or ‘morbid obesity’. It is possible that these terms were defined in line with the definitions given by the World Health Organization [99], but this was not stated.

#### 3.4.2. Demographics

Age was categorised in many ways, making a direct comparison of studies for the Summary of Findings table impractical. The findings of individual studies are all documented in Appendix A). When age was analysed as a continuous variable three of seven studies found that older age correlated with readmission risk, one study found that younger age correlated with increased risk and the remaining three studies found no correlation.

Risk factors correlated with increased readmission risk were: male sex, Black race, decreasing income, low socioeconomic status, Medicare insurance (US-based cohort, private insurance as the reference category), dependent functional status, and frailty. Asian race was protective against readmission.

#### 3.4.3. Other

Patients who underwent elective surgery were less likely to be readmitted than those who underwent urgent surgery, providing the best-available estimation of the impact of the urgency of the patient’s condition on readmission risk. A higher number of previous hospital admissions was correlated with increased readmission risk, whereas a higher number of prior knee procedures was protective against readmission.

The following in-hospital complications that occurred during the index admission, were strongly correlated with readmission risk: any complication (combined category), any medical complication, any surgical complication, urinary tract infection (UTI), surgical site infection (SSI), cardiac complication, pneumonia, acute renal failure. The correlation between in-hospital complication and subsequent risk of 30-day readmission was stronger than for any other variable.

#### 3.4.4. Revision-Only Cohorts

Meta-analysis of 30-day readmission risk for revision TKA was not possible therefore we conducted a narrative synthesis (see Appendix A). Similar to that of the primary TKA studies, in-hospital complications were strongly correlated with readmission after discharge after revision TKA. Of interest, Belmont et al. 2016 [69] and Courtney et al. 2018 [73] reported contrasting findings with respect to the impact of sex on readmission risk, with the former reporting increased risk due to female sex and the latter reporting increased risk due to male sex. Both studies analysed the NSQIP cohort, with Courtney analysing the data from 2012–2016 and Belmont analysing the data from 2011–2012. Unique to this population, revision for infectious aetiology correlated with higher rates of readmission than revision for other indications.

#### 3.4.5. Different Types of Readmission

The focus of this review is on 30-day readmission due to any cause, so the main summary of findings tables comprise studies that analysed all-cause readmission. Three of these studies additionally analysed readmission due to specific causes: D’Apuzzo et al. 2017 [54], Ali et al. 2019 [42], and Rudasil et al. 2019 [89]. An in-depth discussion of the differences between risk factor profiles for each of these outcomes is given in the full text articles. One other study, Anthony et al. 2018 [47], did not analyse all-cause readmission, instead focusing on readmission due to surgical site infection only. On multivariate logistic regression analysis (OR (95% confidence interval)), risk factors found to be associated with risk of readmission were: age 18–30 (reference category <18: 0.114 (0.02–0.63)), female sex (0.559 (0.53–0.59)), private insurance and Medicaid (reference category Medicare: 0.679 (0.63–0.73) and 1.489 (1.32–1.68), respectively), hypertension (1.189 (1.11–1.27)), obesity (1.182 (1.11–1.26)), and diabetes (1.122 (1.05–1.2)). The results for each of these studies are available in Appendix A.

## 4. Discussion

The aims of this systematic review were to (1) identify patient-related characteristics that confer increased risk of unplanned 30-day readmission following TKA and (2) determine the effect size of the association between the identified risk factors and unplanned 30-day readmission.

The risk of readmission within 30 days of TKA is increased in the presence of a broad range of modifiable and non-modifiable patient-related risk factors in the domains of comorbidity, demographics, and socioeconomic status. In relation to the narrative review on this topic [15], the substantial impact of in-hospital complications on readmission risk is a novel finding and the evidence pertaining to the impact of age, BMI, and socioeconomic status was explored in much greater detail. Many of the risk factors analysed comprise patient information routinely collected as part of the work-up patients receive prior to TKA surgery. The findings of this review can therefore be readily used in clinically applicable predictive modelling for 30-day readmission in TKA patients.

In-hospital complications were most strongly associated with 30-day all-cause readmission, for both primary and revision TKA patients. It was beyond the scope of this study to uncover mechanistic prognostic pathways by which these risk factors influence a patient’s likelihood of being readmitted; however, it makes intuitive sense that patients who are susceptible to in-hospital complications prior to discharge may be more susceptible than other patients to complications post-discharge, which lead to hospital readmission. This systematic review presents evidence that in-hospital complications specifically increase the risk of readmission following discharge. This finding was consistent for the broad categories of any complication, any surgical complication, and any medical complication, and also for some specific complications. These included deep vein thrombosis (DVT), pulmonary embolism (PE), urinary tract infection (UTI), surgical site infection (SSI), cardiac complications (including cardiac arrest and myocardial infarction), pneumonia, and acute renal failure. Care is already taken to prevent such complications, for example with the use of prophylactic antibiotics and venous thromboembolism prophylaxis, and patients who experience such complications prior to discharge are already managed with the appropriate measures. These measures include infection control and antimicrobial therapy, venous thromboembolism treatment, renal replacement therapy for acute renal failure, and cardiac interventions and monitoring of cardiac complications. This review provides further evidence that, even when such appropriate measures are taken to prevent and manage these complications, there is still an increased risk of readmission following discharge. As such, even after apparent stabilisation of the patient to such an extent that they are cleared for discharge, it may be prudent to continue monitoring these patients in the post-discharge period to prevent avoidable readmission. This could take the form of a scheduled outpatient or telehealth appointment within the weeks following discharge in order to check in with the patient and assess whether they have any concerns or issues that need to be addressed in order prevent readmission. Further research is required to elucidate the causes and timing of readmission post-TKA such that post-discharge monitoring for patients with in-hospital complications is allocated appropriately and delivered at the appropriate time.

### Strengths and Limitations

The strengths of this study include the comprehensive search strategy and rigorous narrative synthesis and meta-analysis of an extensive range of risk factors, according to a pre-defined and published protocol [20]. Meta-analyses consistently included 100,000 to one million patients, and narrative syntheses of some variables exceeded one million patients. Heterogeneity (measured by the I^2^ statistic) was low in most cases. The findings were synthesised and summarised in accordance with a formulation of the GRADE framework specifically modified to suit prognostic factor systematic reviews.

Age, and to a lesser extent BMI, was categorised in many different ways, as has been previously reported [15], making comparison between studies difficult. There is no universal consensus for categorising age. The best available guidelines [100] recommend categorising age into under 1 y; single-year for 1–24 y; 5-year groupings for 25–54 y; single-year for 55–74 y; 5 year groupings for 75–85; 85 y+. No study in this review adhered to this highest level of detail. Moving forward, we recommend researchers consider utilising this categorisation scheme wherever possible to ensure optimal statistical comparisons of findings between studies.

While we did not place a restriction on the inclusion of prospective studies, 63 of the 69 included studies were retrospective cohort studies. Retrospective studies do not provide the same level of evidence as prospective studies designed specifically to analyse risk factors for readmission. Another issue was that of sample dependence, with 33 of 69 studies included in this review drawing data from the American College of Surgeons National Surgical Quality Improvement Program (NSQIP) database [101]. This posed a challenge for our analysis because studies with overlapping data collection periods needed to be treated as if they were the same study population.

Some studies reported analyses that combined both urgent and elective cases. For example, the comparison of elective TKA with non-elective TKA depicted in Table 6 is arguably flawed as these two patient populations are fundamentally different. These patient populations were not presented separately in the included studies and therefore could not be analysed independently.

Another potential limitation of this review is the lack of account for policy changes aimed at reducing 30-day readmission rates in many of the included studies. The Hospital Readmissions Reduction Program (HRRP) in the USA was expanded to include TKA from 2015 onwards, the UK has had a national readmission reduction program in place since 2011, and Denmark has included public reporting of readmission rates and provided indirect incentives to reduce readmission rates since 2006 [11,102]. In this review, one USA study used data entirely from 2016, 38 USA studies analysed data prior to 2015, and the remaining 18 USA studies (18/57 or 32% of all USA studies) used a data collection window that included 2015 (see Table 3). Only two of these 18 studies adjusted their analyses for year of surgery [89,92], and thus providing some level of control for potential confounding introduced by the expansion of the HRRP program to include TKA in 2015. The remaining 16 studies are potentially limited by the fact that there was no consideration in the analysis for the impact of such a major policy change being introduced during their data collection period. There was only one study from a UK population (see Table 3), which similarly did not adjust for year of surgery nor otherwise account for the fact that the national readmission study was introduced during their data collection period. The two studies from Denmark included in this review analysed data collected after 2006 (see Table 3), the year public reporting of readmission rates commenced in Denmark. To strengthen the evidence for the risk factors included in this review, we call for future prospective cohort studies involving a more diverse range of databases from different geographical locations, taking into account major policy changes in their analyses.

## 5. Conclusions

Thirty-day all-cause readmission risk following TKA is increased in the presence of a broad range of modifiable and non-modifiable patient-related risk factors in the domains of comorbidity, demographics, and low socioeconomic status. The strongest risk factors are in-hospital complications prior to discharge, suggesting patients who suffer a complication could benefit from closer monitoring in the post-discharge period aimed at preventing avoidable readmission. Future research could be conducted to determine whether closer post-discharge monitoring would prevent unplanned but avoidable readmissions. The effects of age and BMI were difficult to analyse due to variations in the categorisation of these variables. Body mass index was not strongly correlated with readmission, while the correlation between older age and readmission was inconsistent. These findings can be used by clinicians when working with patients to reduce their risk of post-TKA readmission. Statisticians can also use these findings in predictive modelling to implement readmission risk reduction systems. A predictive model that utilises sophisticated statistical and machine learning methodologies could draw together the extensive range of variables correlated with readmission that we have identified and harness the underlying complexities of the data to make robust predictions for individual patients in order to personalise care.

## Figures and Tables

**Figure 1 jcm-10-00134-f001:**
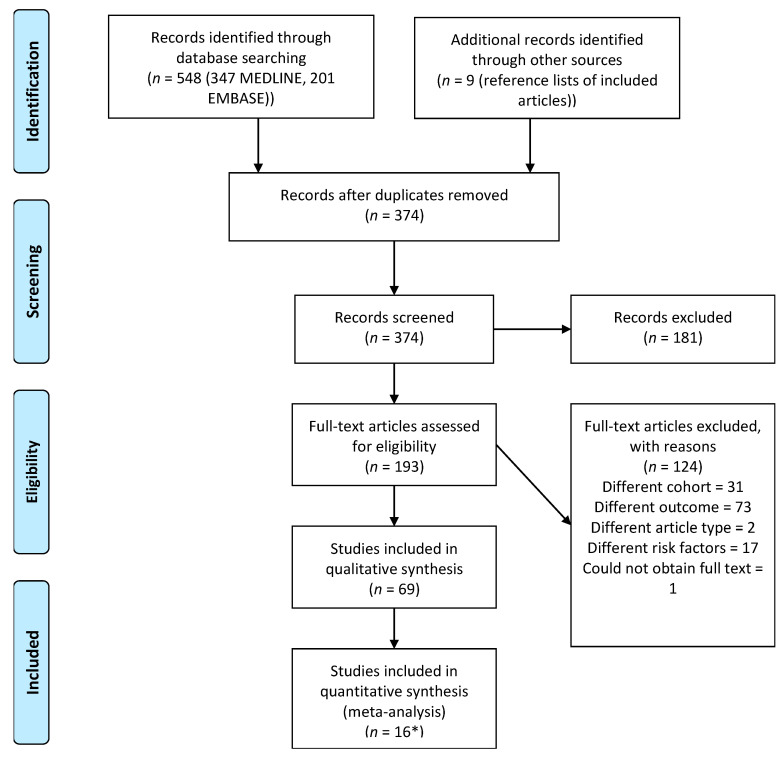
PRISMA Flow Diagram. PRISMA flow diagram depicting the number of articles screened and included in the review. * included in both meta-analysis and qualitative synthesis

**Table 1 jcm-10-00134-t001:** Example of critical appraisal table.

Critical Appraisal												
Study ID	* JBI 1.	JBI 2.	JBI 3.	JBI 4.	JBI 5.	JBI 7.	JBI 8.	JBI 9.	JBI 10.	JBI 11.	Overall Risk of Bias Score (Quartile; Arranged in Descending Order)	Evidence of Selective Reporting
Example study	Y	Y	Y	Y	Y	Y	Y	N	U	Y	3/20 therefore 15% (1st quartile)	No

The colour scheme mimics that of a traffic light system indicating whether each critical appraisal criterion was fulfilled in the given study: Green = yes (Y), Yellow = unclear (U), Red = No (N); * Joanna Briggs Institute critical appraisal checklist for cohort studies – components: Joanna Briggs Institute (JBI) 1. Were the groups similar and recruited from the same population? JBI 2. Were the exposures measured similarly to assign people to both exposed and unexposed groups? JBI 3. Was the exposure measured in a valid and reliable way? JBI 4. Were confounding factors identified? JBI 5. Were strategies to deal with confounding factors stated? JBI 6. (Omitted because it is not relevant) Were the groups/participants free of the outcome at the start of the study (or at the moment of exposure)? JBI 7. Were strategies to deal with confounding factors stated? JBI 8. Was the follow up time reported and sufficient to be long enough for outcomes to occur? JBI 9. Was follow up complete, and if not, were the reasons for loss to follow up described and explored? JBI 10. Were strategies to address incomplete follow up utilized? JBI 11. Was appropriate statistical analysis used?

**Table 2 jcm-10-00134-t002:** Example of summary of findings table.

**Meta-analysis**							
**Prognostic factor**	**OR (95% CI)**		**Overall quality**				
**Example prognostic factor 1**	0.82 (0.71–0.95)		5 (+++)				
**Narrative Synthesis**							
**Prognostic factor**	**Univariate**		**Multivariate**				**Overall quality**
	+	0	-	+	0	-	
**Example prognostic factor 2**	-	-	-	-	1	-	4 (+++)
**Example category 2**							
**Example prognostic factor 3**	-	-	-	2	-	-	5 (+++)

Overall quality: +++ moderate quality = moderately confident in the effect estimate: true effect is likely to be close to the estimate of the effect, but there is a possibility that it is substantially different.

**Table 3 jcm-10-00134-t003:** Study characteristics.

Country	Study ID: Author and Year of Publication (Study Period)
Australia	Hanly 2017 [32] (Retrospective cohort study; January 2003–December 2010)
Canada	Abdulla 2020 [33] (March 2010 to July 2016); Ross 2020 [34] (2003–2016); Peskun 2012 [35] (1997–2007)
Colombia	Buitrago 2020 [36] (January 1, 2012 to November 30, 2015)
Denmark	Jorgensen 2013 [31] (1 February 2010–1 May 2011); Jorgensen 2017 [37] (2012–2014)
Singapore	Tang 2019 [38] (January 2013 to June 2014); Tay 2017 [39] (Jan 2006–Dec 2011)
Taiwan	Liao 2016 [40] (January 1, 2004 to December 31, 2009); Kuo 2017 [41] (January 2009 to 2012)
UK	Ali 2019 [42] (2006–2015)
USA	Miric 2014 [43] (04/01/2001 to 12/31/2011); Welsh 2017 [44] (2009–2011); Kim 2019 [45] (January 1, 2010 to December 31, 2014); Kurtz 2016 [46] (2010–2013); Anthony 2018 [47] (2013 and 2014, inclusive); Urish 2018 [48] (2014); Sodhi and Mont et al. 2019 [49] (1 January 2016 to 30 September 2016); Singh 2013 [50] (2002 fiscal year); Arroyo 2019 [51] (January 2007 to December 2014 for Florida and New York data, January 2007 to December 2011 for California data, January, 2012 to December 2014 for Maryland data); Bullock 2003 [52] (January 1994–June 2000 for bilateral TKA cohort, January 1995–June 2000 for unilateral TKA cohort); Charette 2019 [53] (April 2013–April 2017); D’Apuzzo 2017 [54] (1997–2014); Keeney 2015 [55] (1 January 2006–30 September 2013); Ramos 2014 [56] (2010 and 2011); Ramos 2014 [56] (2010 and 2011); Ricciardi 2017 [57] (January 2010 to December 2014); Saucedo 2014 [58] (2006–2010); Schaeffer 2015 [59] (July 2011–November 2012); Schairer 2014 [6] (2005–2011); Workman 2019 [60] (1 June 2011–1 June 2016); Siracuse 2017 [61] (2006–2011); Weick 2018 [62] (2003–2014); Kheir 2014 [63] (1 July 2009–30 June 2011); Anderson 2020 [64] (2010–2014); Mudumbai 2019 [65] (2011 financial year)
USA and various international sites	Abola 2018 [66] (2012–2014); Alvi 2015 [67] (2005–2011); Antoniak 2020 [68] (2006–2016); Belmont 2016 [69] (2011 and 2012); Bovonratwet 2018 [70] (2005–2015); Bovonratwet 2019 [71] (2005–2016); Bovonratwet 2020 [72] (2012–2017); Courtney 2018 [73] (January 1 2012 to December 31 2015); Curtis 2018 [74] (2008–2014); Curtis 2019 [75] (2012–2016); George 2018 [76] (1 January 2011– 31 December 2015); Gwam 2020 [77] (2008–2016); Hart 2016 [78] (2011–2013); Jauregui 2015 [79] (2011); Kester 2016 [80] (January 1 2010–December 31 2013); Lehtonen 2018 [81] (2012–2015); Lovecchio 2014 [82] (2005–2011); Nowak and Schemitsch 2019 [83] (2005–2016); Ottesen 2018 [84] (2005–2015); Patel 2020 [85] (2011–2017); Patterson 2018 [86] (2005–2015); Pugely 2013 [23] (2011 (whole year + 30 days after 31/12/11)); Robinson 2017 [87] (2012–2014; Roth 2019 [88] (unclear); Rudasill 2019 [89] (2010–2016); Runner 2017 [90] (2005–2014); Sloan 2020 [91] (January 2008 to December 2016); Sodhi and Anis et al. 2019 [92] (2011–2016); Suleiman 2015 [93] (2010–2012); Sutton 2016 [94] (1 January 2011–31 December 2012); Webb 2017 [95] (2005–2014); Yohe 2018 [96] (2008–2014); Zusmanovic 2018 [97] (1 January 2008–31 December 2015)

**Table 4 jcm-10-00134-t004:** Summary of findings–comorbidities.

Meta-analysis								
**Prognostic factor**	**OR (95% CI)**			**Overall quality**				
**Alcohol abuse**	1.08 (0.96–1.20)			6 (++++)				
**BMI < 18.5 vs. normal**	1.15 (0.45–2.98)			4 (+++)				
**BMI 25–30 vs. normal**	0.91 (0.80–1.03)			6 (++++)				
**BMI 30–35 vs. normal**	0.90 (0.80–1.02)			6 (++++)				
**BMI 35–40 vs. normal**	0.84 (0.69–1.02)			6 (++++)				
**BMI > 40 vs. normal**	1.05 (0.84–1.31)			6 (++++)				
**Obesity**	1.06 (1.02–1.09)			4 (+++)				
**Weight loss**	0.95 (0.71–1.27)			3 (++)				
**Arrhythmias and AF combined**	1.14 (1.09–1.19)			4 (+++)				
**Ischaemic heart disease (IHD) + coronary artery disease (CAD) + cardiac disease**	1.29 (0.79–2.09)			3 (++)				
**Peripheral vascular disease**	1.17 (1.10–1.24)			5 (+++)				
**Previous myocardial infarction, or coronary artery disease**	1.18 (0.29–4.88)			3 (++)				
**Non-insulin-dependent diabetes mellitus (NIDDM)**	1.08 (0.80–1.45)			4 (+++)				
**Liver disease**	1.29 (1.20–1.39)			4 (+++)				
**Peptic ulcer disease**	0.94 (0.84–1.07)			4 (+++)				
**Anaemia**	1.19 (1.15–1.24)			6 (++++)				
**Deficiency anaemias**	1.06 (1.01–1.11)			4 (+++)				
**Coagulopathy**	1.25 (1.15–1.36)			4 (+++)				
**Fluid and electrolyte disorder**	1.05 (1.00–1.12)			4 (+++)				
**Chronic pulmonary disease**	1.28 (1.22–1.34)			3 (++)				
**Paralysis**	1.13 (0.97–1.31)			4 (+++)				
**Psychiatric disorder**	1.43 (1.12–1.70)			3 (++)				
**Smoking**	1.25 (0.82–1.91)			5 (+++)				
**Rheumatologic disorder**	1.11 (1.04–1.18)			3 (++)				
**Narrative Synthesis**								
**Prognostic factor**	**Univariate**			**Multivariate**			**Overall quality**	
	+	0	-	+	0	-		
**Composite comorbidity indices**								
**Charlson Comorbidity Index (CCI) 1-2 (reference category = 0)**	-	-	-	2	-	-		6 (++++)
**CCI 1 (reference category = 0)**	-	-	-	2	1	-		6 (++++)
**CCI 2 (reference category = 0)**	-	-	-	1	-	-		5 (++++)
**CCI ≥2 (reference category = 0)**	-	-	-	1	1	-		6 (++++)
**CCI 3-4 (reference category = 0)**	-	-	-	1	-	-		5 (+++)
**CCI ≥3 (reference category = 0)**	-	-	-	1	-	-		4 (+++)
**CCI 5+ (reference category = 0)**	-	-	-	1	-	-		5 (+++)
**Increasing CCI**	-	-	-	-	1	-		3 (++)
**Presence of any comorbidity**	1	-	-	2	-	-		4 (+++)
**Increasing American Society of Anaesthesiologists (ASA) classification (reference category = 2)**	-	-	-	1	-	-		4 (+++)
**Increasing ASA classification (reference category = 1)**	-	-	-	-	1	-		2 (+)
**ASA classification (other)**	-	1	-	-	1	-		2 (+)
**Increasing Elixhauser Index**	-	-	-	1	-	-		5 (+++)
**Increasing DRG (Diagnosis-related group)**	-	-	-	1	-	-		3 (++)
**Cardiovascular**								
**Hypertension**	2	2	-	6	1	-		5 (+++)
**Hyperlipidaemia**	1	1	-	-	-	-		4 (+++)
**Cardiac disease**	-	-	-	1	-	-		3 (++)
**Cardiovascular disease (CVD)**	-	1	-	-	-	-		3 (++)
**Congestive Cardiac/Heart Failure (CCF/CHF)**	3	-	-	5	2	-		5 (+++)
**Valvular disease**	1	-	-	2	1	-		3 (++)
**Peripheral vascular disease**	1	1	-	-	1	-		3 (++)
**History of percutaneous coronary intervention or cardiac surgery**	1	-	-	-	-	-		2 (+)
**BMI, obesity, and weight loss**								
**BMI (continuous)**	-	1	-	1	1	-		4 (+++)
**BMI underweight (reference category = overweight (25–30))**	-	-	-	-	1	-		4 (+++)
**BMI normal weight (reference category = overweight (25–30))**	-	-	-	-	1	-		4 (+++)
**BMI obese (reference category = overweight (25–30))**	-	-	-	-	1	-		4 (+++)
**BMI very obese (reference category = overweight (25–30))**	-	-	-	-	1	-		4 (+++)
**BMI morbidly obese (reference category = overweight (25–30))**	-	-	-	1	-	-		4 (+++)
**BMI > 30 (reference category = normal weight)**	-	-	-	1	-	-		2 (+)
**Increasing BMI (reference category = <25)**	-	1	-	-	-	-		1 (+)
**Obesity**	1	1	-	2	1	-		2 (+)
**Morbid obesity**	-	1	-	-	-	-		2 (+)
**Weight loss**	-	1	-	-	1	-		4 (+++)
**Endocrine**								
**Diabetes (general category and Peskun type two diabetes mellitus (T2DM))**	-	1	-	5	4	-		3 (++)
**Diabetes (with complications)**	1	-	-	1	-	-		5 (+++)
**Diabetes (without complications)**	1	-	-	1	-	-		5 (+++)
**Insulin-dependent diabetes mellitus (IDDM)**	1	-	-	1	-	-		5 (+++)
**NIDDM**	1	-	-	-	1	-		4 (+++)
**Hypothyroidism**	-	1	-	-	1	-		5 (+++)
**Gastrointestinal**								
**Liver disease**	1	-	-	-	1	-		2 (+)
**Haematological**								
**Anaemia**	-	-	-	1	1	1		4 (+++)
**Anaemia (blood loss)**	1	-	-	-	1	-		4 (+++)
**Anaemia (deficiency)**	1	-	-	-	2	-		3 (++)
**Bleeding disorders**	1	-	-	1	-	-		3 (++)
**Coagulopathy**	1	-	-	-	-	-		3 (++)
**Anticoagulant therapy**	-	-	-	-	1	-		2 (+)
**Increasing INR**	1	-	-	1	-	-		4 (+++)
**Fluid and electrolyte disorder**	1	-	-	1	-	-		3 (++)
**Elevated serum blood urea nitrogen (BUN)**	1	-	-	1	-	-		4 (+++)
**Hyponatraemia**	-	1	-	-	1	-		4 (+++)
**Low albumin**	-	-	-	1	-	-		4 (+++)
**Elevated creatinine**	1	-	-	-	-	-		3 (++)
**Elevated WBC count**	1	-	-	-	-	-		3 (++)
**Reduced haematocrit**	1	-	-	-	-	-		3 (++)
**Low platelets**	1	-	-	-	-	-		3 (++)
**Respiratory**								
**Chronic obstructive pulmonary disease (COPD) and chronic airways disease (combined)**	-	-	-	4	-	-		3 (++)
**Pulmonary disease**	1	-	-	1	1	-		4 (+++)
**Smoking**	1	2	-	1	-	-		3 (++)
**Pulmonary circulation disorder**	-	-	-	1	1	-		2 (+)
**Asthma**	-	1	-	-	-	-		1 (+)
**Dyspnoea**	1	-	-	-	1	-		3 (++)
**Previous pneumonia**	-	-	-	-	1	-		3 (++)
**Obstructive sleep apnoea**	-	1	-	-	-	-		1 (+)
**Cardiopulmonary disease**	1	-	-	-	-	-		0 (+)
**Psychiatric**								
**Depression**	1	2	-	3	1	-		5 (+++)
**‘Other’ mental health condition (other than depression)**	-	-	-	1	-	-		3 (++)
**Bipolar disorder**	-	1	-	-	-	-		1 (+)
**Post-traumatic stress disorder (PTSD)**	-	1	-	-	-	-		1 (+)
**Anxiety disorder**	-	2	-	-	-	-		2 (+)
**Alcohol abuse**	1	2	-	-	-	-		3 (++)
**Drug abuse (including general substance abuse designation, and drug/alcohol abuse (combined category in Kurtz))**	1	1	-	2	-	-		4 (+++)
**Psychoses**	1	-	-	-	-	-		2 (+)
**Neoplastic**								
**History of cancer**	1	-	-	2	1	-		4 (+++)
**Disseminated cancer**	1	-	-	1	2	-		3 (++)
**Lymphoma**	-	-	-	2	-	-		4 (+++)
**Neurological**								
**Previous stroke**	-	1	-	1	-	1		4 (+++)
**Dementia**	-	-	-	1	-	-		4 (+++)
**Other neurological disorder**	1	-	-	2	1	-		3 (++)
**In-hospital complications**								
**Deep vein thrombosis**	-	-	-	1	-	-		5 (+++)
**Pulmonary embolism**	-	-	-	1	-	-		5 (+++)
**Any complication**	1	-	-	1	-	-		6 (++++)
**Any medical complication**	1	-	-	1	-	-		4 (+++)
**Any surgical complication**	1	-	-	1	-	-		4 (+++)
**Urinary tract infection**	-	-	-	1	-	-		5 (+++)
**Surgical site infection**	-	-	-	1	-	-		5 (+++)
**Sepsis**	1	-	-	-	1	-		4 (+++)
**Cardiac (including cardiac arrest and myocardial infarction)**	-	-	-	1	-	-		4 (+++)
**Pneumonia**	-	-	-	1	-	-		5 (+++)
**Acute renal failure**	-	-	-	1	-	-		5 (+++)
**Cerebrovascular accident (CVA) or transient ischaemic attach (TIA)**	1	-	-	-	-	-		3 (++)
**Renal**								
**Chronic kidney disease (CKD)**	1	-	-	2	1	-		5 (+++)
**Dialysis dependence**	-	-	-	1	-	-		4 (++)
**Renal failure–acute, preoperative**	-	-	-	-	1	-		4 (++)
**Renal failure/disease–chronicity unspecified**	1	-	-	5	-	-		6 (++++)
**Rheumatological and autoimmune**								
**Rheumatoid arthritis/collagen vascular diseases**	1	-	-	-	-	-		3 (++)
**Steroid or other immunosuppressant use for chronic condition**	-	-	-	1	-	-		3 (++)
**Other**								
**Preoperative opioid use**	-	-	-	2	1	-		3 (++)
**Post-discharge opioid use**	-	-	-	1	-	-		1 (+)
**Preoperative medication use (general)**	-	-	-	1	-	-		3 (++)
**Preoperative medication use (analgesics)**	-	-	-		1	-		1 (+)
**Preoperative medication use (anticonvulsants)**	-	-	-	1	-	-		2 (+)
**Preoperative medication use (Serotonin–norepinephrine reuptake inhibitor (SNRIs))**	-	-	-	-	1	-		2 (+)
**Preoperative medication use (Tricyclic antidepressants (TCAs))**	-	-	-	-	1	-		2 (+)
**Preoperative medication use (sedatives)**	-	-	-	-	1	-		2 (+)
**Wound class**	1	-	-	-	2	-		4 (+++)

Overall quality: + very low quality = very little confidence in the effect estimate: true effect likely to be substantially different from the estimate of effect; ++ low quality = confidence in the effect estimate is limited: the true effect may be substantially different from the estimate of the study; +++ moderate quality = moderately confident in the effect estimate: true effect is likely to be close to the estimate of the effect, but there is a possibility that it is substantially different; ++++ high quality = very confident that the true effect lies close to that of the estimate of the effect.

**Table 5 jcm-10-00134-t005:** Summary of Findings–Demographics.

**Meta-analysis**							
**Prognostic factor**	**OR (95% CI)**			**Overall quality**			
**Hispanic race**	0.92 (0.68–1.25)			5 (+++)			
**Narrative Synthesis**							
**Prognostic factor**	**Univariate**			**Multivariate**			**Overall quality**
	+	0	-	+	0	-	
**Age (continuous variable)**							
**Age**	3	1	-	4	3	-	5 (+++)
**Sex**							
**Female sex**	-	1	-	-	1	4	5 (+++)
**Male sex**	2	2	-	8	-	-	6 (++++)
**Race**							
**Black (reference = white or non-Black)**	1	1	-	4	3	-	4 (+++)
**Hispanic (reference = white or non-Hispanic)**	-	1	-	-	-	-	1 (+)
**Asian (reference = white)**	-	2	-	-	1	2	4 (+++)
**Native Hawaiian (reference = white)**	-	1	-	-	-	-	3 (++)
**American Indian (reference = white)**	-	2	-	-	1	-	4 (+++)
**White**	-	1	1	-	1	-	2 (+)
**Indian (reference = Chinese)**	-	-	-	-	1	-	2 (+)
**Malay (reference = Chinese)**	-	-	-	-	1	-	2 (+)
**Biracial (Workman) or mixed race (Ali)**	1	-	-	1	-	-	5 (+++)
**Minority ethnicity**	-	-	-	1	-	-	2 (+)
**Other (Tang = Chinese; otherwise = white)**	-	1	-	1	4	2	3 (++)
**Missing**	-	1	-	1	-	3	3 (++)
**Race (combined analysis – i.e., racial difference exists between readmitted and non-readmitted cohorts)**	2	-	-	1	-	-	4 (+++)
**Socioeconomic**							
**Decreasing incoming**	-	-	-	3	1	-	5 (+++)
**Low socioeconomic status**	1	-	-	1	-	-	4 (+++)
**Insurance status**							
**Medicare (reference category = private insurance or non-Medicare)**	-	-	-	4	-	-	4 (+++)
**Medicaid (reference category = private insurance)**	-	-	-	3	-	-	3 (++)
**Self-pay, no charge, workers’ compensation, or other (reference category = private insurance)**	-	-	-	-	2	3	2 (+)
**Disability entitlement**	-	-	-	1	-	-	3 (++)
**Functional status, living situation, and frailty**							
**Dependent functional status**	-	-	-	1	-	-	4 (+++)
**Use of walking aids**	1	-	-	-	1	-	4 (+++)
**Living alone**	-	1	-	-	2	-	5 (+++)
**Living in an institution or nursing home**	-	1	-	1	-	-	3 (++)
**Homeless**	-	-	-	1	-	-	2 (+)
**Frailty (Modified Frailty Index)**	-	-	-	1	-	-	5 (+++)

Overall quality: + very low quality = very little confidence in the effect estimate: true effect likely to be substantially different from the estimate of effect; ++ low quality = confidence in the effect estimate is limited: the true effect may be substantially different from the estimate of the study; +++ moderate quality = moderately confident in the effect estimate: true effect is likely to be close to the estimate of the effect, but there is a possibility that it is substantially different; ++++ high quality = very confident that the true effect lies close to that of the estimate of the effect.

**Table 6 jcm-10-00134-t006:** Summary of Findings–Other.

**Meta-analysis**								
**Prognostic factor**	**OR (95% CI)**			**Overall quality**				
**Elective vs non-elective**	0.82 (0.71–0.95)			5 (+++)				
**Narrative Synthesis**								
**Prognostic factor**	**Univariate**				**Multivariate**			**Overall quality**
	+	0	-		+	0	-	
**Operative variables**								
**Elective or non-elective procedure**	-	-	-		-	1	-	4 (+++)
**Emergency procedure**	-	-	-		-	1	-	4 (+++)
**Traumatic indication for TKA**	-	-	-		1	1	-	4 (+++)
**Bilateral procedure**	-	1	-		2	2	-	3 (++)
**Revision surgery (vs primary)**	1	1	-		1	-	-	3 (++)
**Healthcare utilisation**								
**Increasing number of previous admissions**	-	-	-		2	-	-	5 (+++)
**Number of prior knee procedures**	-	-	-		-	-	1	4 (+++)
**GP visit between surgery and readmission**	1	-	-		-	-	1	3 (++)
**Radiation therapy within 90 days prior to procedure**	-	-	-		-	-	-	N/A
**Chemotherapy within 30 days prior to procedure**	-	1	-		-	-	-	2 (+)
**Prior operation**	-	-	-		-	1	-	1 (+)
**>30 outpatient visits in the 365 days prior to procedure**	-	-	-		1	-	-	1 (+)
**Patient-reported outcome measures**	-	-	-		1	-	-	1 (+)
**Patient location**	-	-	-		-	1	1	2 (+)

N/A = not-assessable; Overall quality: + very low quality = very little confidence in the effect estimate: true effect likely to be substantially different from the estimate of effect; ++ low quality = confidence in the effect estimate is limited: the true effect may be substantially different from the estimate of the study; +++ moderate quality = moderately confident in the effect estimate: true effect is likely to be close to the estimate of the effect, but there is a possibility that it is substantially different.

## Data Availability

Data is contained within the article or supplementary material.

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
