# Peer review of "Patient-Related Risk Factors for Unplanned 30-Day Hospital Readmission Following Primary and Revision Total Knee Arthroplasty: A Systematic Review and Meta-Analysis"

_jcm, 2021, doi:10.3390/jcm10010134_

Round 1
Reviewer 1 Report
The authors have performed a meta-analysis to assess and identify patient-related characteristics which are associated with increased risk of unplanned 30-day readmission following total knee arthroplasty. The authors determined that in-hospital complications during the index admission were the strongest risk factor for 30-day readmission in primary and revision TKA patients. The authors further state that the strongest risk factors are in hospital complications prior to discharge, suggesting patients who suffer complications would benefit from closer monitoring and discharge planning.
It is this reviewer’s opinion that the manuscript is not acceptable for publication. The major concerns are listed below.
- Do the authors have sufficient information to report the patients that required readmission were not monitored more closely and had less targeted discharge planning? I believe this is the fundamental flaw of this manuscript.
- The authors report that MEDLINE and EMBASE were searched from inception to February 5, 2020. The long interval of time includes several changes in health care that may or may not be relevant to the authors’ topic. Specifically, it wasn’t until 2009 that CMS introduced the hospital readmission reduction program and further expanded this five years later. The changes by CMS greatly affected the way care was provided to patients and the urgency for short hospital stay.
- The authors also report that patients who underwent elective surgery were less likely to be readmitted than those who underwent urgent surgery. I believe this is a third major flaw as these populations are completely different.
Reviewer 2 Report
Dear authors, thank you very much for submitting this informative paper.
This is a very exciting topic for the arthroplasty community because it is also very much about patient satisfaction after surgery. The chapters Introduction and Methods read very well. In terms of results, I think the very detailed tables are too confusing. These should perhaps be added as a supplement.
In the discussion I miss the clear presentation of the important comborbidities, which are important for a 30-day readmission, e.g. a urinary tract infection. Here, the important hospital task to avoid this infection could be pointed out more clearly.
Reviewer 3 Report
This paper deals with the relationship between patient-specific risk factors and unplanned readmission within 30 days after implantation of a total endoprosthesis.
The authors identified a total of 10 clinical conditions that suggest a short-term readmission of patients
The style, content and structure of the work correspond to the usual scientific procedure.
The chapters Introduction and Methods are very good. In particular, the latter is very detailed and good to understand.
Although it is a very interesting paper, some question remain:
- It would be good in the Chapter Methods to list the original MESHs that were used for searching (2.3). The risk of bias etc. (2.6, 2.8) could better contain a small table to explain the checklist.
- It is unclear, what means … multiple studies were analysed…. (line 186).
- Why did the authors show all studies in 3.1? (table1 for example). They couldn´t be summarized?
- I tried to find a connection between the results of 3.3 Risk of bias… and the attending tables, but I didn´t understand. On one side, the authors describe extentd theire results usning tables, on the other side they write, that the exclusion of non-English studies did not affect the findings. Where from did they conclude that?
- Furthermore I didn´t understand, which of all diseases did lead to the conclusion, that for example congestive heart failure is a risk factor.
- The discussion other than the abstract describes first age, BMI and socioeconomic status as risk factors.
- I couldn´t find in the discussion and the conclusion all the results, described in the abstract.
- Literature includes all necessary papers.
Round 2
Reviewer 3 Report
The current text is written correctly from the point of view of the reviewers. All the critical points were essentially taken into account. The tables are still far too long. The last paragraph of the discussion on age and BMI (line 465-471) should have to be moved forward, semanticly, the discussion ends with line 463. There are two self-citations (15, 20), but one of them (20) describes the Project, this is correct. After correcting this parts, from the perspective of the reviewer, the work can be published.
